# Synthesis and Fluorescent Properties of Aminopyridines and the Application in “Click and Probing”

**DOI:** 10.3390/molecules27051596

**Published:** 2022-02-28

**Authors:** Zongyang Li, Yaxuan Li, Wenxu Chang, Sen Pang, Xuefeng Li, Liusheng Duan, Zhenhua Zhang

**Affiliations:** 1College of Agronomy and Biotechnology, China Agricultural University, Beijing 100193, China; lizongy@126.com (Z.L.); 2018310060128xuan@cau.edu.cn (Y.L.); changwx@cau.edu.cn (W.C.); pangsen7812@cau.edu.cn (S.P.); 91030@cau.edu.cn (X.L.); duanlsh@cau.edu.cn (L.D.); 2College of Science, China Agricultural University, Beijing 100193, China

**Keywords:** azide, aminopyridine, fluorescence, click-and-probing

## Abstract

Unsubstituted pyridin-2-amine has a high quantum yield and is a potential scaffold for a fluorescent probe. However, the facile access to conjugated highly substituted aminopyridines and the study of their fluorescent properties is scarce. In this paper, synthesis and fluorescent properties of multisubstituted aminopyridines were studied based on a recently developed Rh-catalyzed coupling of vinyl azide with isonitrile to form a vinyl carbodiimide intermediate, following tandem cyclization with an alkyne. An aminopyridine substituted with an azide group as a potential probe was further designed, synthesized, and evaluated. The “clicking-and-probing” experiment of it on BSA protein showed the potential of aminopyridine as a scaffold of a biological probe.

## 1. Introduction

The fluorescent molecules have been widely applied in various areas including analyzing proteins, immunoassays, visual recognition, efficient isolation of cells, etc. [1,2,3,4,5]. The selective labeling of such molecules onto proteins or other biomolecules draws attention, accompanied with the development of bioorthogonal chemistry, including Staudinger ligation and click chemistry, which also enables molecules with fluorescence to be applied as reporter groups [6,7,8]. In particular, some compounds with a certain functional group, the azido group for instance, exhibit a “pre-fluorescence”, in that the florescence is switched on only when the click reaction happens between azide and alkyne. Through the “clicking-and-probing” ligation protocol employing the molecules with pre-fluorescence, the issue of background fluorescence and the impractical washing procedure that usually occurred in traditional bioorthogonal chemistry is largely addressed [9,10,11,12]. In order that the fluorescent molecules could be better applied in the detection and imaging of biological signaling, improvements when designing new molecules should be focused on their simplicity, high selectivity, and sensitivity. The scope of the fluorescent compound used in “click and probing” ligation strategy is usually restricted to the derivations of coumarin, fluorescein, and FITC, leading to the limited amount of the matched building blocks [13,14,15].

Unsubstituted pyridin-2-amine with a high quantum yield (*Φ* = 0.6) is a potential scaffold for a fluorescent probe, since it is small in size, which enables it to be taken into the biotic environment with less hinderance [16,17]. In recent years, several cyclization routes to substituted 2-aminopyridines have been reported [18,19,20,21,22,23,24,25,26]. However, the facile access to conjugated highly substituted aminopyridines and studies of their fluorescent properties are scarce. No application has shown aminopyridines as one of the bioorthogonally activated fluorescent probes.

Herein, we synthesized a series of conjugated multisubstituted aminopyridines based on our facile Rh-catalyzed reaction of vinyl azide with isonitrile to form a vinyl carbodiimide intermediate following tandem cyclization with an alkyne [25] and investigated their fluorescent properties. Substituent manipulation (including alkyl versus aryl and methyl ester versus ethyl ester) of the aminopyridines scaffold at positions 3, 4, 6, and 2-amine was realized, which crucially increases the structure diversity and the associated optical properties. Thus, we designed and synthesized one aminopyridine equipped with an azide group, which had no fluorescence, to be a potential probe applied in a clicking-and-probing protocol conjugated to alkynes (Figure 1). The good quantum yield of fluorescence, more suitable maximum emission wavelength, and successful fluorescent labeling of bovine serum albumin (BSA) showed their potential application as biological probes.

## 2. Results and Discussion

### 2.1. Synthesis and Fluorescent Properties of Multisubstituted Aminopyridines

Multisubstituted aminopyridines were synthesized following our pre-published method. Briefly, Rh-catalyst and ligand were dissolved in 1,4-dioxane (2 mL). The vinyl azide and isonitrile were added by syringe after the tube was sealed under N_2_ atmosphere and the reaction mixture was stirred at rt. After the spot of vinyl azide disappeared on TLC, NH_4_Cl, NaHCO_3_, and alkyne were added and the mixture was heated to 120 °C for 8 h. The reaction solution was concentrated in vacuum, and the residue was chromatographed to afford the product without further recrystallization [25]. In addition, the solution of amino pyridine was diluted to 10 μM for quantitative fluorescent detection [27]. The results were listed in Table 1.

We first investigated 2-amino-6-phenylpyridine-3,4-dicarboxylates that contain different substituents on the amine group. They showed no variance in absorbance, excitation, and emit wavelength (λ_A_ = 270 nm, λ_ex_ = 390 nm, λ_em_ = 480 nm) when the substituent group was tertiary butyl (**1**), benzyl group (**2**), and cyclohexyl (**3**) as the quantum yield were 0.34, 0.44, and 0.31, respectively. Furthermore, when the R^1^ group was altered to 4-(trifluoromethyl)phenyl (**4**), 4-(methyl)phenyl (**5**), or 2-(methyl)phenyl (**6**), the emit wavelength exhibited a redshift of 5 nm compared to aminopyridine **1**. Meanwhile, there was only a slight fluctuation around 0.3 in their quantum yield. When the substitution on the benzene was bromine at the para-position (**7**), its quantum yield distinctly decreased to 0.22 as the emit wavelength remained the same as other substituted phenyl group. In addition, aminopyridine **8** with n-octyl at position 6 showed a low quantum yield (0.02) and distinctly decreased in characteristic wavelengths compared with **1**, which has a phenyl. The absorbance wavelength of **8** went down to 258 nm from 270 nm, and it could be excited maximum with the light of 345 nm when the maximum emission wavelength was 455 nm, which was about 30 nm shorter than products with an aromatic group. At the same time, aminopyridines bearing methyl ester (**9**) and (2-methoxyethyl) ester (**10**) showed identical characteristic wavelength with the one having an ethyl ester; the quantum yield of these two compounds is also near 0.3.

### 2.2. Synthesis and Fluorescent Properties of 6-Phenyl Substituted Aminopyridine Derivatives

With diethyl 2-(*tert*-butylamino)-6-phenylpyridine-3,4-dicarboxylate (**1**) in hand, the functional groups of which were transformed as follows (Figure 2): (1) The tertiary butyl group of **1** was cleaved using CF_3_COOH to furnish compound **11**, which showed no fluorescence. The free amino group could be further transformed to other functional groups. (2) The ester groups were hydrolyzed to carboxylic acid to afford compound **12**, which still had a good quantum yield (0.31). (3) Compound **1** was reduced to a high *Φ* (0.81) compound **13** using LiAlH_4_ in excellent yield, but the λ_em_ was reduced to 400 nm.

### 2.3. Synthesis and Fluorescent Properties of Pre-Fluorescence Aminopyridine

As shown in Figure 3, the BocNH-substituted aminopyridine **14** was first synthesized under the standard conditions shown in Section 2.1. Then, the azido substituted aminopyridine **15** was synthesized via a Sandmeyer reaction, and it had no fluorescence because of the quenching effect from the electron-rich nitrogen of the azido group. With catalytic Cu(II), the azide aminopyridine **15** could reacted smoothly with phenylacetylene or propynol in a mixed solution of ethanol and water (1:1) to afford triazole products **16** or **17** at room temperature. Aminopyridines **16** and **17** both fluoresce at 480 nm because of the elimination of the quenching through the formation of the triazole ring from the azido group [11,13]. In comparison to 0.03 of the starting materials **15**, the quantum yield of two products, **16** and **17**, is 0.35 and 0.43, respectively. The cycloaddition of triazole-substituted aminopyridine **17** underwent significant fluorescence enhancement of 0.03 to 0.45 upon click reaction at a level that was 14 times as high as its starting azido aminopyridine **15**.

### 2.4. Kinetic Experiments of the Click Reaction of Pre-Fluorescence Aminopyridine

The efficiency of the click reaction is a determining factor during the bio-application [28,29,30,31,32]. As the acceleration of rate of the cycloaddition may be achieved in various conditions, the rate of CuAAC between pre-fluorescence azido aminopyridine **15** and propargyl alcohol was measured, and the result showed in Figure 4. Reactions were performed with 50 μM **15**, 500 μM propargyl alcohol, 500 μM sodium ascorbate, 50 μM CuSO_4_, and 100 μM ligands. These reactions were carried out in a mixture of EtOH (DMF) and H_2_O solutions. The fluorescence of triazole product **17** were measured every minute during a total time course of 35 min.

Six conditions were evaluated. Condition A: no ligand, EtOH/H_2_O = 1:1; Condition B: TBTA (100 μM), EtOH/H_2_O = 1:1; Condition C: THPTA (100 μM), EtOH/H_2_O = 1:1; Condition D: TBTA (100 μM), DMF/H_2_O = 20:80; Condition E: THPTA (100 μM), DMF/H_2_O = 20:80; and Condition F: THPTA (100 μM), DMF/H_2_O = 5:95. (TBTA = tris[(1-benzyl-1H-1,2,3-triazol-4-yl)methyl]amine, THPTA = tris(3-hydroxypropyltriazolylmethyl)amine).

Among these conditions, TBTA in 50% H_2_O with 50% EtOH (condition B) showed the highest activity in accelerating the cycloaddition as the fluorescent intensity reached 1200 with 3 min, followed by THPTA in 80% H_2_O with 20% DMF (condition E). Notably, only after 10 min did the fluorescence intensity of the latter tend to go up comparing to other conditions. Besides, when the proportion of DMF and H_2_O changed to 5% and 95% (condition F), a slight acceleration could be seen, indicating that the mixture of DMF and H_2_O is a better solvent for THPTA as ligand in comparison with H_2_O and EtOH (condition C). Unfortunately, we could hardly see the difference between TBTA in the mixed solvent with 20% DMF (condition D) and the controlled condition with no ligand.

### 2.5. In Vitro “Clicking and Probing” on Conjugated BSA

Based on the above results, we further evaluated whether the newly designed fluorescent probe could be applied to a real biological environment for the purpose of labeling. A clicking-and-probing experiment was carried out between aminopyridines and BSA conjugated with an alkynyl group.

As shown in Figure 5, BSA was conjugated with a terminal alkyne as follows: BSA (10 mg/mL) was incubated with 1 mM TCEP (TCEP = Tris(2-carboxyethyl)phosphine) in PBS (PBS = phosphate buffered saline) buffer at room temperature for 30 min. Then 1-(prop-2-yn-1-yl)-1H-pyrrole-2,5-dione was added to the solution at a final concentration of 1 mM, and the whole solution was incubated at room temperature for 4 h. Five volumes of methanol were added to the solution to precipitate the protein, and the mixture was then centrifuged at 10,000 rpm for 10 min. The supernatant was discarded, and the precipitate was washed three times with methanol, and then redissolved in PBS buffer containing 2% SDS.

Thus, we detected the fluorescence of the in vitro click reaction. Click reactions were performed in a 20 μL system with 40 μg above conjugated BSA, 50–500 mM probe **15**, 10 mM sodium ascorbate, 200 μM CuSO_4_, and 200 μM indicated ligands (TBTA/THPTA/BTTES/BTTAA), respectively. Reaction mixtures were incubated and shaken gently at room temperature for 1 h. For in-gel fluorescence detection, the reaction mixture was mixed with 5X loading buffer and directly loaded on SDS gel for electrophoresis. Gels were analyzed with a ChemiDoc^TM^ MP system (Biorad) using a 530/28 filter. After fluorescence detection, the gel was stained with Coomassie Blue to indicate the protein loading amount (Figure 6). In-gel fluorescence detection indicated that the labeling efficiency of compound **15** increased in a concentration-dependent manner, and 200 μM gave a best result (for more details, see Appendix A). Also, all of the four commonly used ligands, TBTA, THPTA, BTTES, and BTTAA, could facilitate the clicking-and-probing labeling process. Among them, BTTES showed the lowest activity in click reaction, followed by TBTA and THPTA, with BTTAA exhibiting the highest activity.

## 3. Materials and Methods

### 3.1. General

All reactions were performed in a Schlenk reaction flask. All solvents were purchased from Sinopharm Chemical Reagent (Beijing, China), and THF was redistilled by sodium. The boiling point of petroleum ether is between 60 and 90 °C. For chromatography, 200–300 mesh silica gel (Qingdao, China) was employed. All Rh catalysts and BSA (A1933) were purchased from Sigma-Aldrich (Shanghai, China). All ligands and substrates were purchased from Energy Chemical. 

^1^H and ^13^C NMR spectra were recorded at 400 MHz and 100 MHz with Varian Mercury 400 spectrometer ((Agilent Technologies, Palo Alto, CA, USA) at ambient temperature. Chemical shifts are reported in ppm relative to chloroform (^1^H, δ 7.26; ^13^C, δ 77.00), DMSO (^1^H, δ 2.50; ^13^C, δ 39.52). IR spectra were recorded with a Nicolet AVATAR 330 FT-IR spectrometer (Thermo Fisher Scientific, Madison, WI, USA). UV absorbance and fluorescence were measured on a PerkinElmer Lambada 650S (PerkinElmer Inc., Waltham, MA, USA) and HITACHI F-4500 (Hitachi Ltd., Tokyo, Japan), respectively. Mass spectra were obtained on a Waters Auto Purification LC/MS system (Waters Corp., Milford, MA, USA). HMRS were obtained on a Bruker Apex IV FTMS spectrometer (Bruker Corp., Karlsruhe, Germany). Kinetic experiments were performed using a 96-well BioTek Synergy Hybrid Plate Reader and carried out by the EnspireTM 2300 Multilabel Reader (PerkinElmer Inc., Waltham, MA, USA). In-gel fluorescence was detected by a Bio-Rad ChemiDocTM MP system (Bio-Rad Laboratories, Hercules, CA, USA) using a 530/28 filter.

### 3.2. Experimental Procedures for the Synthesis of ***1**–**17***

In a 5 mL tube, catalyst [Rh(COD)Cl]_2_ (0.005 mmol) and ligand 2,2′-bpy (0.01 mmol) were dissolved in 1,4-dioxane (2 mL). Vinyl azide (0.2 mmol) and isonitrile (0.2 mmol) were added by syringe after the tube was sealed under N_2_ atmosphere and the reaction mixture was stirred at rt. After the spot of vinyl azide disappeared on TLC, NH_4_Cl (0.2 mmol), NaHCO_3_ (0.2 mmol), and alkyne (0.4 mmol) were added and the mixture was heated to 120 °C for 8 h. The reaction solution was concentrated in vacuum, and the residue was chromatographed with petroleum and ethyl acetate as eluent to afford aminopyridine products **1**–**10** and **14**.

In a 5 mL tube, **1** (0.2 mmol) and CF_3_OOH (0.4 mmol) were dissolved in PhMe (2 mL). The mixture was stirred and refluxed until **1** disappeared, which was judged by TLC. The reaction solution was concentrated in vacuum, and the residue was chromatographed with petroleum and ethyl acetate as eluent (PE/EA = 3:1) to afford **11**.

In a 5 mL tube under N_2_ atmosphere, **1** (0.20 mmol) and *t*-BuOK (0.80 mmol) were added in THF (2 mL). The mixture was stirred at rt until **1** disappeared, which was judged by TLC. The solution pH was adjusted to 3 by HCl (3 M). Then the mixture partitioned between EtOAc and H_2_O, and the organic phase was separated, dried (Na_2_SO_4_), and concentrated in vacuum. The residue was chromatographed with petroleum and ethyl acetate as eluent (PE/EA = 1:1) to afford **12**.

In a 5 mL tube under N_2_ atmosphere, **1** (0.20 mmol) was dissolved in THF (2 mL), and then LiAlH_4_ (0.4 mmol) was added slowly in an ice bath. The mixture was stirred at rt until **1** disappeared, which was judged by TLC. The solution pH was adjusted to 7 by HCl (3 M). Then the mixture partitioned between EtOAc and H_2_O, and the organic phase was separated, dried (Na_2_SO_4_) and concentrated in vacuum. The residue was chromatographed with petroleum and ethyl acetate as eluent (PE/EA = 3:1) to afford **13**.

In a 5 mL tube under N_2_ atmosphere, **14** (0.20 mmol) was dissolved in a mixed solution of THF (1 mL) and water (1 mL), and then conc. HCl (0.5 mL) was added slowly in an ice bath. The mixture was stirred at rt until **14** disappeared, which was judged by TLC. Then NaN_3_ (0.24 mmol) was added and the reaction mixture was stirred at rt for another 2h. Then the mixture partitioned between EtOAc and H_2_O, and the organic phase was separated, dried (Na_2_SO_4_) and concentrated in vacuum. The residue was chromatographed with petroleum and ethyl acetate as eluent (PE/EA = 20:1) to afford **15**.

In a 5 mL tube, catalyst CuSO_4_ (0.005 mmol) and NaAsc (0.01 mmol) were dissolved in a mixed solution of EtOH (1 mL) and water (1 mL). Then, azido aminopyridine **15** (0.2 mmol) and alkyne (0.24 mmol) were added and the reaction mixture was stirred at rt until **15** disappeared, which was judged by TLC. Then the mixture was partitioned between EtOAc and H_2_O, and the organic phase was separated, dried (Na_2_SO_4_), and concentrated in vacuum. The residue was chromatographed with petroleum and ethyl acetate as eluent (PE/EA = 10:1) to afford **16** and **17**.

### 3.3. Characterization Details 

Diethyl 2-(*tert*-butylamino)-6-phenylpyridine-3,4-dicarboxylate (**1**). Light yellow solid. ^1^H NMR (400 MHz, CDCl_3_) δ 8.08–8.03 (m, 2H), 7.97 (s, 1H), 7.50–7.42 (m, 3H), 6.98 (s, 1H), 4.36 (q, *J* = 7.2 Hz, 2H), 4.30 (q, *J* = 7.1 Hz, 2H), 1.57 (s, 9H), 1.38 (t, *J* = 7.2 Hz, 3H), 1.34 (t, *J* = 7.2 Hz, 3H). ^13^C NMR (100 MHz, CDCl_3_) δ 169.05, 167.06, 159.24, 157.85, 145.95, 138.51, 130.00, 128.76, 127.36, 105.81, 101.13, 61.82, 61.42, 51.85, 29.31, 14.27, 14.14. HRMS (ESI+), *m/z* [M + H]^+^, calcd for C_21_H_27_N_2_O_4_: 371.1971, found: 371.1959. λ_A_ = 270 nm, λ_ex_ = 390 nm, λ_em_ = 480 nm, *Φ* = 0.34.

Diethyl 2-(benzylamino)-6-phenylpyridine-3,4-dicarboxylate (**2**). Light yellow solid. ^1^H NMR (400 MHz, CDCl_3_) δ 8.27 (s, 1H), 8.02–7.97 (m, 2H), 7.46–7.37 (m, 5H), 7.36–7.30 (m, 2H), 7.28–7.23 (m, 1H), 7.02 (s, 1H), 4.87 (d, *J* = 5.6 Hz, 2H), 4.38 (q, *J* = 7.2 Hz, 2H), 4.31 (q, *J* = 7.2 Hz, 2H), 1.39 (t, *J* = 7.2 Hz, 3H), 1.33 (t, *J* = 7.1 Hz, 3H). ^13^C NMR (100 MHz, CDCl_3_) δ 168.88, 166.73, 159.86, 157.83, 145.99, 139.79, 138.15, 130.17, 128.71, 128.64, 127.65, 127.42, 127.12, 106.86, 101.20, 61.92, 61.51, 45.28, 14.26, 14.12. HRMS (ESI+), *m/z* [M + H]^+^, calcd for C_24_H_25_N_2_O_4_: 405.1814, found: 405.1797. λ_A_ = 270 nm, λ_ex_ = 390 nm, λ_em_ = 480 nm, *Φ* = 0.31.

Diethyl 2-(cyclohexylamino)-6-phenylpyridine-3,4-dicarboxylate (**3**). Light yellow solid. ^1^H NMR (400 MHz, CDCl_3_) δ 8.08–8.00 (m, 2H), 7.89 (d, *J* = 7.3 Hz, 1H), 7.50–7.41 (m, 3H), 6.95 (s, 1H), 4.37 (q, *J* = 7.2 Hz, 2H), 4.31 (q, *J* = 7.2 Hz, 2H), 2.13–2.03 (m, 2H), 1.82–1.73 (m, 2H), 1.69–1.59 (m, 1H), 1.52–1.45 (m, 2H), 1.44–1.21 (m, 10H). ^13^C NMR (100 MHz, CDCl_3_) δ 169.07, 166.81, 159.84, 157.44, 146.05, 138.38, 130.08, 128.73, 127.33, 105.98, 100.48, 61.84, 61.35, 49.42, 32.99, 26.08, 24.94, 14.26, 14.16. HRMS (ESI+), *m/z* [M + H]^+^, calcd for C_23_H_29_N_2_O_4_: 397.2127, found: 397.2112. λ_A_ = 270 nm, λ_ex_ = 390 nm, λ_em_ = 480 nm, *Φ* = 0.44.

Diethyl 2-(*tert*-butylamino)-6-(4-(trifluoromethyl)phenyl)pyridine-3,4-dicarboxylate (**4**). Light yellow solid. ^1^H NMR (400 MHz, CDCl_3_) δ 8.16 (d, *J* = 8.2 Hz, 2H), 7.90 (s, 1H), 7.71 (d, *J* = 8.1 Hz, 2H), 6.99 (s, 1H), 4.42–4.29 (m, 4H), 3.63 (q, *J* = 6.7 Hz, 2H), 1.72–1.63 (m, 2H), 1.52–1.43 (m, 2H), 1.43–1.32 (m, 6H), 0.98 (t, *J* = 7.3 Hz, 3H). ^13^C NMR (101 MHz, CDCl_3_) δ 168.71, 166.70, 158.17, 158.12, 146.12, 141.68, 131.67 (q, *J* = 32.5 Hz), 127.65, 125.68 (q, *J* = 3.8 Hz), 124.21 (q, *J* = 544.5, 272.2 Hz), 106.43, 101.87, 62.02, 61.61, 41.11, 31.64, 20.46, 14.27, 14.14, 14.07. HRMS (ESI+), *m/z* [M + H]^+^, calcd for C_22_H_26_F_3_N_2_O_4_: 489.1845, found: 489.1845. λ_A_ = 270 nm, λ_ex_ = 390 nm, λ_em_ = 485 nm, *Φ* = 0.31.

Diethyl 2-(*tert*-butylamino)-6-(p-tolyl)pyridine-3,4-dicarboxylate (**5**). Light yellow solid. ^1^H NMR (400 MHz, CDCl_3_) δ 7.99–7.91 (m, 3H), 7.32–7.21 (m, 2H), 6.94 (s, 1H), 4.42–4.23 (m, 4H), 2.41 (s, 3H), 1.56 (s, 9H), 1.42–1.30 (m, 6H). ^13^C NMR (100 MHz, CDCl_3_) δ 169.17, 167.10, 159.30, 157.88, 145.90, 140.25, 135.79, 129.51, 127.32, 105.53, 100.73, 61.80, 61.36, 51.82, 29.32, 21.52, 14.27, 14.16. HRMS (ESI+), *m/z* [M + H]^+^, calcd for C_22_H_29_N_2_O_4_: 385.2127, found: 385.2108. λ_A_ = 270 nm, λ_ex_ = 390 nm, λ_em_ = 485 nm, *Φ* = 0.27.

Diethyl 2-(*tert*-butylamino)-6-(o-tolyl)pyridine-3,4-dicarboxylate (**6**). Light yellow solid. ^1^H NMR (400 MHz, CDCl_3_) δ 7.94 (s, 1H), 7.43–7.40 (m, 1H), 7.31–7.28 (m, 1H), 7.27–7.24 (m, 2H), 6.58 (s, 1H), 4.37–4.27 (m, 4H), 2.41 (s, 3H), 1.47 (s, 9H), 1.39–1.31 (m, 6H). ^13^C NMR (100 MHz, CDCl_3_) δ 168.89, 167.20, 162.79, 157.71, 145.06, 140.33, 135.82, 130.79, 129.57, 128.63, 125.80, 109.94, 100.71, 61.81, 61.48, 51.87, 29.46, 20.72, 14.26, 14.14. HRMS (ESI+), *m/z* [M + H]^+^, calcd for C_22_H_29_N_2_O_4_: 385.2127, found: 385.2114. λ_A_ = 270 nm, λ_ex_ = 390 nm, λ_em_ = 485 nm, *Φ* = 0.32.

Diethyl 2-(*tert*-butylamino)-6-(4-bromophenyl)pyridine-3,4-dicarboxylate (**7**). Light yellow solid. ^1^H NMR (400 MHz, CDCl_3_) δ 7.97 (s, 1H), 7.93–7.89 (m, 2H), 7.61–7.57 (m, 2H), 6.93 (s, 1H), 4.36 (q, *J* = 7.3 Hz, 2H), 4.30 (q, *J* = 7.3 Hz, 2H), 1.55 (s, 9H), 1.38 (t, *J* = 7.2 Hz, 3H), 1.34 (t, *J* = 7.1 Hz, 3H). ^13^C NMR (75 MHz, CDCl_3_) δ 168.82, 166.98, 158.09, 157.82, 146.11, 137.52, 131.98, 128.91, 124.56, 105.63, 101.79, 61.88, 61.51, 51.90, 29.29, 14.26, 14.12. HRMS (ESI+), *m/z* [M + H]^+^, calcd for C_21_H_26_BrN_2_O_4_: 449.1076, found: 449.1051. λ_A_ = 270 nm, λ_ex_ = 390 nm, λ_em_ = 485 nm, *Φ* = 0.22.

Diethyl 2-(*tert*-butylamino)-6-octylpyridine-3,4-dicarboxylate (**8**). Light yellow solid. ^1^H NMR (400 MHz, CDCl_3_) δ 7.87 (s, 1H), 6.30 (s, 1H), 4.47–4.12 (m, 4H), 2.71–2.57 (m, 2H), 1.82–1.66 (m, 2H), 1.48 (s, 9H), 1.39–1.18 (m, 16H), 0.87 (t, *J* = 6.9 Hz, 3H). ^13^C NMR (100 MHz, CDCl_3_) δ 169.20, 167.18, 166.30, 157.94, 145.17, 108.27, 99.57, 61.65, 61.19, 51.75, 38.59, 32.00, 29.59, 29.41, 29.35, 29.33, 29.29, 28.62, 22.81, 14.24, 14.14. HRMS (ESI+), *m/z* [M + H]^+^, calcd for C_23_H_39_N_2_O_4_: 407.2904, found: 407.2900. λ_A_ = 258 nm, λ_ex_ = 345 nm, λ_em_ = 455 nm, *Φ* = 0.02.

Dimethyl 2-(cyclohexylamino)-6-phenylpyridine-3,4-dicarboxylate (**9**). Light yellow solid. ^1^H NMR (400 MHz, CDCl_3_) δ 8.12–7.97 (m, 2H), 7.82 (d, *J* = 7.2 Hz, 1H), 7.57–7.37 (m, 3H), 6.97 (s, 1H), 4.33–4.15 (m, 1H), 3.91 (s, 3H), 3.84 (s, 3H), 2.13–2.02 (m, 2H), 1.88–1.70 (m, 2H), 1.54–1.30 (m, 6H). ^13^C NMR (100 MHz, CDCl_3_) δ 169.52, 167.16, 160.02, 157.38, 145.70, 138.28, 130.17, 128.76, 127.34, 105.95, 100.43, 52.79, 52.32, 49.48, 32.98, 26.07, 24.94. HRMS (ESI+), *m/z* [M + H]^+^, calcd for C_21_H_25_N_2_O_4_: 369.1814, found: 369.1799. λ_A_ = 270 nm, λ_ex_ = 390 nm, λ_em_ = 480 nm, *Φ* = 0.35.

Bis(2-methoxyethyl) 2-(*tert*-butylamino)-6-phenylpyridine-3,4-dicarboxylate (**10**). Light yellow solid. ^1^H NMR (400 MHz, CDCl_3_) δ 8.04 (dd, *J* = 7.7, 1.9 Hz, 2H), 7.82 (s, 1H), 7.45 (m, 3H), 7.01 (s, 1H), 4.64–4.23 (m, 4H), 3.84–3.57 (m, 4H), 3.39 (d, *J* = 3.4 Hz, 6H), 1.56 (s, 9H). ^13^C NMR (100 MHz, CDCl_3_) δ 168.92, 166.93, 159.34, 157.71, 145.66, 138.48, 130.03, 128.76, 127.39, 106.10, 101.31, 70.31, 70.21, 65.03, 64.41, 59.20, 58.95, 51.91, 29.31. HRMS (ESI+), *m/z* [M + H]^+^, calcd for C_23_H_31_N_2_O_6_: 431.2177, found: 431.2164. λ_A_ = 270 nm, λ_ex_ = 390 nm, λ_em_ = 480 nm, *Φ* = 0.30.

Diethyl 2-amino-6-phenylpyridine-3,4-dicarboxylate (**11**). Light yellow solid. ^1^H NMR (400 MHz, CDCl_3_) δ 8.35 (s, 2H), 7.98–7.77 (m, 2H), 7.67–7.42 (m, 3H), 6.96 (s, 1H), 4.52–4.25 (m, 4H), 1.50–1.31 (m, 6H). ^13^C NMR (100 MHz, CDCl_3_) δ 166.96, 164.79, 157.46, 157.19, 148.73, 133.59, 131.82, 129.26, 128.04, 108.88, 104.33, 62.64, 62.51, 14.17, 14.05. HRMS (ESI+), *m/z* [M + H]^+^, calcd for C_17_H_19_N_2_O_4_: 315.1345, found: 315.1332. λ_A_ = 270 nm, λ_ex_ = 390 nm, λ_em_ = 480 nm, *Φ* = 0.01.

2-(*tert*-Butylamino)-6-phenylpyridine-3,4-dicarboxylic acid (**12**). Light yellow solid. ^1^H NMR (400 MHz, DMSO-*d6*) δ 13.39 (s, 2H), 8.27–8.00 (m, 3H), 7.65–7.36 (m, 3H), 7.17 (s, 1H), 1.53 (s, 9H). ^13^C NMR (100 MHz, DMSO-*d6*) δ 169.47, 168.42, 157.49, 157.25, 147.70, 137.80, 130.04, 128.89, 126.82, 105.18, 102.11, 51.03, 28.87. HRMS (ESI+), *m/z* [M + H]^+^ calcd for C_17_H_19_N_2_O_4_: 315.1345, found: 315.1333. λ_A_ = 270 nm, λ_ex_ = 390 nm, λ_em_ = 480 nm, *Φ* = 0.30.

(2-(*tert*-Butylamino)-6-phenylpyridine-3,4-diyl)dimethanol (**13**). Light yellow solid. ^1^H NMR (400 MHz, CDCl_3_) δ 8.03–7.97 (m, 2H), 7.47–7.38 (m, 2H), 7.38–7.32 (m, 1H), 6.92 (s, 1H), 4.46 (d, *J* = 11.7 Hz, 4H), 1.54 (s, 9H). ^13^C NMR (100 MHz, CDCl_3_) δ 157.93, 154.40, 147.28, 139.66, 128.66, 128.61, 126.64, 115.72, 108.20, 63.11, 57.75, 51.58, 29.63. HRMS (ESI+), *m/z* [M + H]^+^, calcd for C_17_H_23_N_2_O_2_: 287.1760, found: 287.1748. λ_A_ = 330 nm, λ_ex_ = 335 nm, λ_em_ = 400 nm, *Φ* = 0.81.

Diethyl 6-(4-((*tert*-butoxycarbonyl)amino)phenyl)-2-(cyclohexylamino)pyridine-3,4-dicarboxylate (**14**). Light yellow solid. ^1^H NMR (400 MHz, CDCl_3_) δ 7.99 (dd, *J* = 8.6, 1.4 Hz, 2H), 7.89 (d, *J* = 7.2 Hz, 1H), 7.45 (d, *J* = 7.8 Hz, 2H), 6.90 (s, 1H), 6.65 (s, 1H), 4.36 (q, *J* = 7.2 Hz, 2H), 4.30 (q, *J* = 7.1 Hz, 2H), 4.25–4.16 (m, 1H), 2.15–2.03 (m, 2H), 1.84–1.70 (m, 2H), 1.71–1.60 (m, 2H), 1.54 (s, 8H), 1.45–1.29 (m, 10H). ^13^C NMR (100 MHz, CDCl_3_) δ 169.93, 169.21, 166.83, 159.20, 157.42, 152.56, 146.02, 140.28, 132.87, 128.23, 118.19, 105.33, 61.83, 61.27, 49.48, 32.98, 28.45, 27.04, 26.10, 24.97, 14.26, 14.18. HRMS (ESI+), *m/z* [M + H]^+^, calcd for C_28_H_38_N_3_O_6_: 512.2755, found: 512.2750. λ_A_ = 270 nm, λ_ex_ = 390 nm, λ_em_ = 480 nm, *Φ* = 0.67.

Diethyl 6-(4-azidophenyl)-2-(cyclohexylamino)pyridine-3,4-dicarboxylate (**15**). Light yellow solid. ^1^H NMR (400 MHz, CDCl_3_) δ 8.05 (d, *J* = 8.0 Hz, 2H), 7.90 (d, *J* = 7.4 Hz, 1H), 7.11 (d, *J* = 7.9 Hz, 2H), 6.90 (s, 1H), 4.41–4.27 (m, 4H), 4.24–4.18 (m, 1H), 2.15–2.03 (m, 2H), 1.82–1.73 (m, 2H), 1.69–1.61 (m, 2H), 1.50–1.42 (m, 2H), 1.42–1.32 (m, 8H). ^13^C NMR (100 MHz, CDCl_3_) δ 169.01, 166.75, 158.63, 157.41, 146.15, 141.84, 135.13, 128.86, 119.33, 105.53, 61.90, 61.40, 49.49, 32.96, 26.08, 24.95, 14.27, 14.17. HRMS (ESI+), *m/z* [M + H]^+^, calcd for C_23_H_28_N_5_O_4_: 438.2136, found: 438.2205. λ_A_ = 270 nm, λ_ex_ = 390 nm, λ_em_ = 480 nm, *Φ* = 0.03.

Diethyl 2-(cyclohexylamino)-6-(4-(4-phenyl-1H-1,2,3-triazol-1-yl)phenyl)pyridine-3,4-dicarboxylate (**16**). Light yellow solid. ^1^H NMR (400 MHz, CDCl_3_) δ 8.26 (s, 1H), 8.22 (d, *J* = 8.8 Hz, 2H), 7.96–7.89 (m, 4H), 7.53–7.45 (m, 2H), 7.42–7.35 (m, 1H), 6.99 (s, 1H), 4.43–4.29 (m, 4H), 4.27–4.20 (m, 1H), 2.15–2.06 (m, 2H), 1.85–1.75 (m, 2H), 1.71–1.62 (m, 2H), 1.54–1.43 (m, 2H), 1.43–1.33 (m, 6H), 1.30–1.24 (m, 2H). ^13^C NMR (100 MHz, CDCl_3_) δ 168.85, 166.69, 158.05, 157.40, 148.70, 146.27, 138.71, 138.04, 130.25, 129.10, 128.77, 128.67, 126.03, 120.47, 117.47, 105.92, 101.28, 62.00, 61.54, 49.59, 32.94, 26.07, 24.95, 14.28, 14.16. HRMS (ESI+), *m/z* [M + H]^+^, calcd for C_31_H_34_N_5_O_4_: 540.2605, found: 540.2598. λ_A_ = 255 nm, λ_ex_ = 390 nm, λ_em_ = 480 nm, *Φ* = 0.35.

Diethyl 2-(cyclohexylamino)-6-(4-(4-(hydroxymethyl)-1H-1,2,3-triazol-1-yl)phenyl)pyridine-3,4-dicarboxylate (**17**). Light yellow solid. ^1^H NMR (400 MHz, CDCl_3_) δ 8.20 (d, *J* = 7.2 Hz, 2H), 8.05 (s, 1H), 7.91 (d, *J* = 7.1 Hz, 1H), 7.84 (d, *J* = 7.2 Hz, 2H), 6.98 (s, 1H), 4.92 (s, 2H), 4.44–4.29 (m, 4H), 4.27–4.20 (m, 1H), 2.17–2.02 (m, 2H), 1.86–1.65 (m, 4H), 1.56–1.42 (m, 8H), 1.44–1.30 (m, 8H). ^13^C NMR (100 MHz, CDCl_3_) δ 168.84, 166.68, 158.01, 157.40, 146.27, 138.83, 137.97, 128.79, 120.57, 105.94, 101.32, 62.01, 61.55, 56.88, 49.57, 32.93, 26.06, 24.93, 14.28, 14.16. HRMS (ESI+), *m/z* [M + H]^+^, calcd for C_26_H_32_N_5_O_5_: 494.2398, found: 494.2392. λ_A_ = 280 nm, λ_ex_ = 390 nm, λ_em_ = 480 nm, *Φ* = 0.45. (for details, see Appendix A).

## 4. Conclusions

We synthesized a series of fluorescent molecules with an aminopyridine scaffold and investigated their fluorescent properties. The majority of obtained compounds exhibit good fluorescence characteristics, among which **2** with a cyclohexyl substituting the amine group has the highest quantum yield over 0.40, and its characteristic wavelength meets the requirements as a fluorescent molecule. By means of derivatizing, we designed a bioorthogonally activated smart probe that is nonfluorescent and generates a highly fluorescent fluorophore after a click-labeling reaction with alkynes. This represents a 14-fold change in fluorescence intensity. The efficiency of this “turn on” and “turn off” strategy is also investigated, indicating the reaction could be conducted rapidly with the optimized condition. Hence, the biochemical application of this probe was demonstrated by the BSA protein conjugated “clicking-and-probing” experiment. Therefore, the fluorescent molecules based on multisubstituted aminopyridines have a potential prospect in extensive application of biochemical detection and analysis.

## Data Availability

The data presented in this study are available in Appendix A.

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
