# Peer review of "Synthesis and Fluorescent Properties of Aminopyridines and the Application in “Click and Probing”"

_molecules, 2022, doi:10.3390/molecules27051596_

Round 1

Reviewer 1 Report

In this article, Li and coworkers describe the synthesis and characterization of a series of highly-fluorescent aminopyridines which have been are prepared using a previously developed synthetic strategy. In the case of one of these derivatives (compound 15), it was conjugated with an azido group which functionalization resulted in a strong fluorescence quenching. Finally, click chemistry of the latter to ethynyl-functionalized molecules or bovine serum albumin (BSA) restored the fluorescence thus proving the utility of these molecules as fluorescent tags.

I reckon that this paper offers an interesting example towards the preparation of fluorophores that could be used as labels and that could work, in principle, also in biological environments.

The paper is clear, the experiments well-presented, and the references appropriate so that I recommend its publication in Molecules after addressing the comments/realizing the corrections reported below.

Something that results strange to this referee is that a large number of derivatives have been prepared and characterized in the manuscript but only one of them (compound 15) has been tested in the “click and probing” approach. Then, what is the reason to have prepared and characterized all the other derivatives? What the preparation of the other compounds add to the paper? The authors are invited to comment on that.

Several typos are present in the manuscript which should be carefully checked. Here few examples: haave, serveral, Figure 1 fluorescent spelled wrong twice, isonitirile (twice), External tandard, with a aminopyridines (it is an amino…), detaction, etc.

Pages 8-14. The spectra should be moved to the Supporting information.

Title: I suggest to modify it to “Synthesis and Fluorescent Properties of Aminopyridines and their Application in “Click and Probing” Strategies”

Figures 1, 2, 3 (this could be a Figure but I suggest to change it to Scheme - see my comment below), and 5 are not Figures but Schemes.

Figure 3. I suggest to move the emission spectra to the Supporting and reference to it in the text.

Figure 1: What does it mean in the caption “This is a figure. Schemes follow the same formatting”? It has no meaning.  

“…showed poor quantum yield…” better “…showed low quantum yield…”, “…which remained a good quantum yield…” better “…which showed a good quantum yield …”

“…the standard condition as shown in 2.1.”. What does it mean as shown in 2.1? What is it?

“…due to the elimination of the quenching through the formation of the triazole ring.” Why the presence of an azide results in the quenching of the fluorescence? The authors should comment on that and include some references supporting that.

“…fluorescence of triazole product 16 were measured…” In Figure 4 it seems that the product is 17.

Figure 4. The different conditions should be included in the caption and not only in the text.

I do not like the use of “Pre-fluorescence” in several parts in the manuscript. It does not mean much. I suggest to use something like no-fluorescent or fluorescence-quenched.

Isometric solvent. What is that? Never heard about it.

Figure 6: In the figure it appears compound 11 but in the manuscript it seems that it is 15.

References: Refs 7 (name of the journal in italic), 16 (title not in italics), 27 (something is missing in the brackets), 32 (an “A” is missing in the title of the paper, and journal should be italic, year bold, volume italic).

Acronyms are introduced but their exact meaning is not specified (BSA, TCEP, etc)

Supporting Information: The solvent used in all the NMR spectra should be specified in the captions.

Author Response

1) Something that results strange to this referee is that a large number of derivatives have been prepared and characterized in the manuscript but only one of them (compound 15) has been tested in the “click and probing” approach. Then, what is the reason to have prepared and characterized all the other derivatives? What the preparation of the other compounds add to the paper? The authors are invited to comment on that.

Reply: Thank you for reviewer’s question. With different substituted group, the pyridin-2-amine will haave different fluoresent properties, including absorbance wavelength, excitation wavelength, emit wavelength, and quantum yield. Before the study of “click and probing” of pyridin-2-amine, we need to figure out how the substituted group affect the fluoresent properties (substituent manipulation (including alkyl versus aryl and methyl ester versus ethyl es-ter) of the aminopyridines scaffold at positions 3, 4, 6 and 2-amine). The result of these derivatives supported the design of probe 15.

2) Several typos are present in the manuscript which should be carefully checked. Here few examples: haave, serveral, Figure 1 fluorescent spelled wrong twice, isonitirile (twice), External tandard, with a aminopyridines (it is an amino…), detaction, etc.

Reply: Thanks for reviewer’s suggestion. We’ve carefully checked the manuscript. The updates have been shown in revised-manuscript with highlighted.

3) Pages 8-14. The spectra should be moved to the Supporting information.

Reply: Thank you for reviewer’s suggestion. We’ve moved the sepctra to SI.

4) Title: I suggest to modify it to “Synthesis and Fluorescent Properties of Aminopyridines and their Application in “Click and Probing” Strategies”

Reply: Thank you for reviewer’s suggestion. As this manuscript showed that the fluorescent molecules based on multisubstituted aminopyridines have a potential prospect in extensive application of biochemical detaction and analysis, but did not show a lot of different detailed applications. The previous title would be better.

5) Figures 1, 2, 3 (this could be a Figure but I suggest to change it to Scheme - see my comment below), and 5 are not Figures but Schemes.

Reply: Thank you for reviewer’s suggestion. I’ve changed all “Figure” in this manuscript to “Scheme”

6) Figure 3. I suggest to move the emission spectra to the Supporting and reference to it in the text.

Reply: Thank you for review’s suggestion. In this Figure/Scheme, Compounds 15 and 17 are the keys in this manuscript. It’s necessary to show the emission spectra but not only the value.

7) Figure 1: What does it mean in the caption “This is a figure. Schemes follow the same formatting”? It has no meaning.

Reply: Thank you for reviewer’s question. It’s a mistake. We’ve correct it to “The appliaction of Aminopyridine in Click &Probing”.

8) “…showed poor quantum yield…” better “…showed low quantum yield…”, “…which remained a good quantum yield…” better “…which showed a good quantum yield …”

Reply: Thank you for reviewer’s suggestion. We’ve changed the description.

9) “…the standard condition as shown in 2.1.”. What does it mean as shown in 2.1? What is it?

Reply: Thank you for reviewer’s question. It means the following condition shown in 2.1 (Page 2)

“Briefly, Rh-catalyst and ligand were dissolved in 1,4-dioxane (2 mL). The vinyl azide and isonitirile were added by syringe after the tube was sealed and exchanged N2 atmosphere, the reaction mixture was stirred at rt. After the spot of vinyl azide disappeared on TLC, the NH4Cl, NaHCO3 and alkyne were added and the mixture was heated to 120 oC for 8 h. The reaction solution was concentrated in vacuum, the residue was chromatographed to afford the product without further recrystallization.”

10) “…due to the elimination of the quenching through the formation of the triazole ring.” Why the presence of an azide results in the quenching of the fluorescence? The authors should comment on that and include some references supporting that.

Reply: Thank you for reviewer’s suggestion. In this paragraph, we’ve described that “the azido substituted aminopyridine 15 had no fluorescence due to the quenching effect from the electron-rich nitrogen of the azido group”. The formation of the triazole ring from the azido group eliminated the quenching. We’ve modified the description and added references.

11) “…fluorescence of triazole product 16 were measured…” In Figure 4 it seems that the product is 17.

Reply: Thank you for reviewer’s suggestion. We’ve corrected this mistake.

12) Figure 4. The different conditions should be included in the caption and not only in the text.

Reply: Thank you for reviewer’s suggestion. However, the text about the description of different conditions followed the Figure. It’s not nesscessary to reply it twice in a same page.

13) I do not like the use of “Pre-fluorescence” in several parts in the manuscript. It does not mean much. I suggest to use something like no-fluorescent or fluorescence-quenched.

Reply: Thank you for reviewer’s suggestion. In click and probing, the compounds with certain functional group, the azido group for instance, are fuorescence precursors. After the click reaction, it will have fluorescence. “Pre-fluorescence” is to show this meaning.

14) Isometric solvent. What is that? Never heard about it.

Reply: Thank you for reviewer’s suggestion. We’ve modified it to “TBTA in 50% H2O with 50% EtOH (condition B)”.

15) Figure 6: In the figure it appears compound 11 but in the manuscript it seems that it is 15.

Reply: Thank you for reviewer’s suggestion. We’ve corrected this mistake.

16) References: Refs 7 (name of the journal in italic), 16 (title not in italics), 27 (something is missing in the brackets), 32 (an “A” is missing in the title of the paper, and journal should be italic, year bold, volume italic).

Reply: Thank you for reviewer’s suggestion. We’ve corrected the mistakes.

17) Acronyms are introduced but their exact meaning is not specified (BSA, TCEP, etc)

Reply: Thank you for reviewer’s suggestion. The expanation of BSA has already shown in Page 2 paragraph 2, “bovine serum albumin (BSA)”. We’ve added the explanation of other acronyms.

18) Supporting Information: The solvent used in all the NMR spectra should be specified in the captions.

Reply: Thank you for reviewer’s suggestion. We’ve added the solvent information into SI.

Reviewer 2 Report

The authors report on the synthesis and characterisation of a range of substituted 2-aminopyridine compounds. Substitutions are on the amine group as well as the 3- and 4- and 6- positions of the pyridine ring. The synthesis is achieved using a Rhodium catalysed method developed by the authors, although this is not referenced in the manuscript. Some subsequent functional group interconversion is also undertaken on the 3- and 4- substituents. Characterisation is focussed on fluorescence properties of these molecules in preparation for bioconjugation. One particular substituted derivative has its functionality converted into an azide and monitored for its reaction with an alkyne with a view to subsequent Click reaction with alkyne substituted BSA. Successful conjugation with BSA was observed paving the way for this molecule to be used in Click and Probe type experiments.

The manuscript needs considerable work before being suitable for publication:

  • The manuscript contains a large amount of spelling and grammatical errors
  • Figure 1 needs a proper caption. In addition, substituents need to be added to the aminopyridine ring at the 3 and 4- positions accurately reflecting the molecules used.
  • A reference needs to be added for the synthetic method used
  • The yields for the compounds need to be added into Table 1. Also yields for the reactions in Figure 2 and the final reaction in Figure 3.
  • The NMR spectral data needs correcting. Specifically, the reciprocity in coupling constants. Further, if spectra were run on a 400 MHz spectrometer then the 13C frequency will be 100 MHz. In addition, compound 10 reports a dd for an aromatic proton signal which is difficult to see from where it originates.
  • A comment about the purity of the compounds needs to be added. From the NMR spectra it is clear that some compounds are perhaps only 85% pure. The compounds have been used directly from chromatography column and not recrystalised.
  • The expected HRMS values should be added to allow straightforward comparison with experimentally observed values.
  • Redundant statements at the end of the document should be deleted.
  • Reference 32 need correct formatting.

Author Response

1) The manuscript contains a large amount of spelling and grammatical errors.

Reply: Thanks for reviewer’s suggestion. We’ve carefully checked the manuscript. The updates have been shown in revised-manuscript with highlighted.

2) Figure 1 needs a proper caption. In addition, substituents need to be added to the aminopyridine ring at the 3 and 4- positions accurately reflecting the molecules used.

Reply: Thank you for reviewer’s suggestion. It’s a mistake. We’ve correct it to “The appliaction of Aminopyridine in Click &Probing”. And the 3 and 4- position substitutents have been added.

3) reference needs to be added for the synthetic method used

Reply: Thank you for reviewer’s suggestion. Ref 25 has added to 2.1 paragraph 1.

4) The yields for the compounds need to be added into Table 1. Also yields for the reactions in Figure 2 and the final reaction in Figure 3.

Reply: Thank you for reviewer’s suggestion. The isolated yields have all been added.

5) The NMR spectral data needs correcting. Specifically, the reciprocity in coupling constants. Further, if spectra were run on a 400 MHz spectrometer then the 13C frequency will be 100 MHz. In addition, compound 10 reports a dd for an aromatic proton signal which is difficult to see from where it originates.

Reply: Thank you for reviewer’s suggestion. All 101 MHz has been corrected to 100 MHz. And “7.45 (dd, J = 6.0, 4.8 Hz, 3H)” in compound 10 has been corrected to “7.45 (m, 3H)”. In addtion, we’ve added the solvent information in all the NMR spectra into SI.

6) A comment about the purity of the compounds needs to be added. From the NMR spectra it is clear that some compounds are perhaps only 85% pure. The compounds have been used directly from chromatography column and not recrystalised.

Reply: Thank you for reviewer’s question. We’ve check the NMR spectra again. The CNMR is clear. And it’s clear in 4.0-10.0 ppm in HNMR, especially in aromatic area. The inpure peaks are almost in 1.0-1.5 ppm. The main impure may be the impure of chromatography column solvent, petroleum ether, which has no fluorescence. It would not disturb the study of fluorescence properties. In addition, the amount of every product was only around 50-100 mg. It’s difficult to do the recrystallization. Following reviewer’s suggestion, “without further recrystallization” was added to 2.1 paragraph 1.

7) The expected HRMS values should be added to allow straightforward comparison with experimentally observed values.

Reply: Thank you for reviewer’s suggestion. The expected HRMS values have been added.

8) Redundant statements at the end of the document should be deleted.

Reply: Thank you for reviewer’s suggestion. Redundant statements have been deleted.

9) Reference 32 need correct formatting.

Thank you for reviewer’s suggestion. We’ve corrected the mistakes.